# FIKA: A Conformal Geometric Algebra Approach to a Fast Inverse Kinematics Algorithm for an Anthropomorphic Robotic Arm

**Oscar Carbajal-Espinosa [1,*], Leobardo Campos-Macías [2]** and **Miriam Díaz-Rodriguez [3]**

1 Tecnologico de Monterrey, School of Engineering and Science, Zapopan 45201, Mexico
2 Intelligent Systems Research Laboratory, Intel Corporation, Zapopan 45017, Mexico
3 Unidad Académica Zapopan, Instituto Tecnológico José Mario Molina Pasquel y Henríquez, Tecnológico Nacional de México, Zapopan 45019, Mexico; miriam.diaz@zapopan.tecmm.edu.mx
* Correspondence: oscar.carbajal@tec.mx

**Abstract:** This paper presents a geometric approach to solve the inverse kinematics for an anthropomorphic robotic arm with seven degrees of freedom (DoF). The proposal is based on conformal geometric algebra (CGA), by which many geometric primitives can be operated naturally and directly. CGA allows for the intersection of geometric entities such as two or more spheres or a plane's projection over a sphere. Rigid transformations of such geometric entities are performed using only one operation through another geometric entity called a motor. CGA imposes geometric restrictions on the inverse kinematics solution, which avoids computation of the forward kinematics or other numerical solutions, unlike traditional approaches. Comparisons with state-of-the-art algorithms are included to prove our algorithm's superior performance: such as decreased execution time and errors of the end-effector for a series of desired poses.

**Keywords:** kinematics; humanoid robots; redundant robots

## 1. Introduction

The robotic arm is the most-used robot configuration to approach the human arm's performance. This kind of robotic arm is a multi-articulated rigid limb with many degrees of freedom (DoF) that provide flexibility and agility, allowing a single task to be executed using various postures and different trajectories. For this, the solution of the inverse kinematics (IK) has become one of the techniques for this task, which is the problem of determining an appropriate joint configuration so that the end-effector moves to a desired target position.

A human arm's basic configuration is described as 3-DoF at the shoulder, 1-DoF at the elbow, and 3-DoF at the wrist. Thus, a 7-DoF robotic arm can interact naturally in a designed human environment. One advantage of this configuration is that the elbow creates a self-motion manifold on a circular path, avoiding joint limitations and singularities. For these reasons, it is desired to have the same design in a humanoid robot.

One of the main difficulties when working with this kind of configuration is that there may be many or infinite appropriate joint configurations (which is the common case), a unique solution, or no solution for the same task, i.e., a set of solutions may exist to carry the hand (end-effector) from an initial pose to a final one. This problem is known as redundancy and makes the inverse kinematics more complex than solving it for kinematics chains with lower DoFs for which redundancy is not present.

Many approaches tackle the redundancy problem, and in general, analytical and numerical methods are the most-used. Other techniques use geometric and trigonometric solutions to take advantage of the strong geometric structure representation of a rigid robot arm and combine the classical representation of direct kinematics to solve the problem.

For instance, the main issue of the analytical method is that the computations of the joints become difficult to find when the kinematic chain becomes larger due to the complex system of equations that results in an unfeasible algorithm for real-time implementation. Regarding numerical solutions, they are based on iterative methods for which a cost function has to be minimized.

The algorithm proposed here is based on the logic that a geometric method facilitates the intuition to pose a solution to the problem but is formulated on the conformal geometric algebra (CGA) mathematical framework. This algebra allows working with geometric primitives such as circles, lines, planes, and spheres naturally. These geometric entities can be represented in this algebra as linear combinations of its base, where parts of its base are the canonical base of the Euclidean vector space $\mathbb{R}^3$, and also, it includes rigid transformations defined in terms of geometric entities called rotors, translators, and motors. Using the algebra characteristics, as the rigid transformation of geometric entities, it is possible to define an algorithm that solves the inverse kinematics problem without using matrices or numerical methods. The resultant algorithm allows us to impose geometric constraints based on the physical possibilities of the arm.

Although our previous work [1] involves the inverse kinematics for a leg with 6-DoF using the same mathematical framework, the extension to 7-DoF through the use of CGA is not trivial given the redundancy caused by one extra degree of freedom, for which we had to define more geometric entities to find one extra appropriate joint angle.

Furthermore, extensive research can be found related to the problem of solving the inverse kinematics of redundant robotic arms [2–5]. For example, in the works [6,7], the solution of analytical inverse kinematics for a 7-DoF robotic arm are given in such form that some restrictions to avoid joint limits are included, and many complex operations must be solved along with a stage of forward kinematics, increasing the computational burden of the algorithm. In [8], many matrix multiplications are needed to find a solution for the problem. However, as in the previous two related papers, the authors define a geometric restriction using a rotating plane. Still, given that matrix algebra is not suited to work efficiently with this kind of entity, it is natural to find a mathematical framework to solve this problem. Other approaches to the solution of inverse kinematics in redundant robotic arms are given in the works [9,10], wherein the solution is given using the Jacobian matrix. Its inverse is obtained using numerical methods to optimize the solution of the inverse kinematics. Also, extended approaches can be found, such as [11]. However, this method also poses a pseudo-inverse-like problem; this is in contrast with the purpose of our method, wherein analytical constraints are exploited to produce solutions in a geometrical manner and, as the comparisons demonstrate, with less computational time.

Unlike previous works, our method has the advantage that the use of analytical or numerical solutions is not needed to find appropriate joint angles given a desired pose of the end-effector. This work is an extension of our previous work [12], which solves the inverse kinematics for a 5-DoF manipulator and which was then extended to a 6-DoF leg [1] of a humanoid robot. These results were used as inspiration for this algorithm for a 7-DoF arm, but many improvements are included. In summary, our contributions concerning the state-of-the-art are:

- We propose an algorithm for the inverse kinematics for a 7-DoF robotic arm on a 5D conformal geometric algebra CGA instead of working only on the 3D vector space $\mathbb{R}^3$. The intuitive representation of geometric entities in CGA and the operations that we can make of them allow us to find the joint positions (in Cartesian space) and joint angles of the robot arm by taking into account the outer and inner product that CGA possesses, which avoids analytical or numerical methods. The geometry entities available in the arm structure are used to define CGA entities, e.g., circles are determined to find elbow position. Thus, all the possible configurations for the arm are in those circles, which takes advantage of the dexterity.
- We provide a comparison with state-of-the-art algorithms to demonstrate the superior performance of our proposal.

The rest of the paper is structured as follows: Section 2 provides a brief introduction to CGA. Section 3 presents the proposed method to solve the inverse kinematics, including images that clarify the steps of the algorithm. The comparison with the state-of-the-art algorithms is shown in Section 4. Finally, some discussions are presented in Section 6.

The principal aim of this research work is to develop an inverse kinematics algorithm using conformal geometric algebra that allows us to extend the previous work we have been doing to redundant robotic arms with seven degrees of freedom. With our proposal, the calculation time is reduced by exploiting the properties of the mathematical framework with which we are working. By using this algorithm, we hope to improve the performance of this type of manipulator arm in real time.

Comparisons with other algorithms show that conformal geometric algebra has advantages over other mathematical systems; we take advantage of these differences in our proposal. Unlike other approaches, for which it is necessary to use direct kinematics to find a possible solution to the configuration of the robotic arm, our algorithm uses geometric constraints to resolve the inverse kinematics. Furthermore, the geometric entities defined in the inverse kinematics solution can be used to find other configurations that allow us to take advantage of the redundancy of the robotic arm.

## 2. Mathematical Background

### 2.1. Geometric Algebra

Geometric algebra (GA), also known as Clifford algebra, was introduced by William K. Clifford (1845–1879) and recently has been increasingly applied in physics and engineering. GA is a mathematical framework that allows for treating of the geometry of objects as algebraic entities.

Let $V^{p,q,r}$ be a vector space over the field $\mathbb{R}$. We can generate a geometric algebra $G_{p,q,r}$ from $V^{p,q,r}$, where $p$, $q$, and $r$ stand for the number of basis vectors that square to $1$, $-1$, and $0$, respectively: that is, there are $p, q, r \geq 0$ with $n = p + q + r$ such that

$$e_i e_i = e_i^2 = \begin{cases} 1 & \text{for } i = 1, \ldots, p, \\ -1 & \text{for } i = p+1, \ldots, p+q, \\ 0 & \text{for } i = p+q+1, \ldots, n. \end{cases} \tag{1}$$

The geometric algebra $G_{p,q,r}$ has a base given by these elements:

$$\{1\}, \{e_i\}, \{e_i \wedge e_j\}, \{e_i \wedge e_j \wedge e_k\}, \ldots, \{e_1 \wedge e_2 \wedge \cdots \wedge e_n\}, \tag{2}$$

where

$$I_n := e_1 \wedge e_2 \wedge \cdots \wedge e_n \tag{3}$$

is called as the pseudoescalar.

So $M \in G_{p,q,r}$ (called a multivector) is expressed in terms of its base elements: namely,

$$M = <A>_0 + <A>_1 + \cdots + <A>_n, \tag{4}$$

where each $<A>_j$, $j = 0, \ldots, n$ is the $j$-vector part.

For instance, the Euclidean geometric algebra $G_3 := G_{3,0,0}$ has a base

$$\{1, e_1, e_2, e_3, e_{12}, e_{31}, e_{23}, I_3 = e_{123}\}, \tag{5}$$

where $e_{ij} := e_i \wedge e_j$, and any multivector $M$ can be written in the form

$$M = \alpha_0 + \alpha_1 e_1 + \alpha_2 e_2 + \alpha_3 e_3 + \alpha_4 e_{12} + \alpha_5 e_{31} + \alpha_6 e_{23} + \alpha_7 I_3, \tag{6}$$

$\alpha_i \in \mathbb{R}$, $i = 0, \ldots, 7$.

The main product of $G_{p,q,r}$ is called the geometric product; it is associative and distributive over addition but not necessarily commutative.

The geometric product of two multivectors $A, B \in G_{p,q,r}$ can be written as the sum of its anticommutaror and commutator parts:

$$AB = \frac{1}{2}(AB + BA) + \frac{1}{2}(AB - BA) \tag{7}$$

Moreover, if

$$M = a_1 a_2 \cdots a_{n-1} a_n \tag{8}$$

for some $\{a_1, a_2, \ldots, a_{n-1}, a_n\} \subset V^{p,q,r}$, then it is called a *k*-versor. A *k*-blade is a *k*-versor of orthogonal vectors, and the linear combination of *k*-blades give rise to a *k*-vector.

In particular, the geometric product of two vectors $a$, $b \in \mathbb{R}$ is given by the sum of a symmetric and antisymmetric part:

$$ab = a \cdot b + a \wedge b, \tag{9}$$

where $a \cdot b$ and $a \wedge b$ are the inner product and wedge product, respectively. The inner product of two 1-vectors is the standard cross product in the Euclidean vector space.

The inner product operator is used for the computations of the angles between lines, planes, etc.; the wedge product is mainly used for the construction and intersection of geometric entities, while the geometric product is used for the description of transformations.

### 2.2. Conformal Geometric Algebra

It is well known that objects, such as points, lines, circles, spheres, etc., can be described in $\mathbb{R}^3$. However, in this vector space, we cannot make an operation on these objects. Then, it is natural to think of how to make this possible. Then, we should use a 5D geometric algebra: for this, take an orthonormal base for $\mathbb{R}^3$, say $\{e_i\}$, $i = 1, 2, 3$, and join it with this orthonormal base $\{e_4, e_5\}$, which is the base of a Minkowski plane such that

$$e_4^2 = 1, \; e_5^2 = -1, \; e_4 \cdot e_5 = 0. \tag{10}$$

In this case, we have $p = 4$, $q = 1$, $r = 0$ as in (1); then, we have $\mathbb{R}^{4,1} := \mathbb{R}^{4,1,0}$, which gives $G_{4,1} := G_{4,1,0}$. This GA is called *conformal geometric algebra* (CGA), by which geometric entities such as points, lines, and spheres can be represented as vectors that have two algebraic representations: the inner product null space (IPNS) and the outer product null space (OPNS).

Also, CGA has the sphere as its unity of calculus; this means that the representation of the other geometric primitives such as lines, points, planes, circles, and point pairs can be obtained from the representation of the sphere, as we will see in the next subsection.

From the Minkowski plane base, $\{e_4, e_5\}$ gives rise to another base that has these elements:

$$e_\infty = e_5 + e_4, \quad e_0 = \frac{1}{2}(e_5 - e_4), \tag{11}$$

where $e_\infty$ and $e_0$ are the point at infinity and the origin point, respectively.

These new vector bases satisfy

$$e_\infty^2 = e_0^2 = 0; \tag{12}$$

that is, $e_\infty$ and $e_0$ are null vectors, and also, their inner product satisfies

$$e_\infty \cdot e_0 = -1. \tag{13}$$

Often the null vectors are used to represent the geometric entities instead of $e_4$ and $e_5$. In the base of $G_3$ given in (5), we have the pseudo-scalar

$$I_3 = e_1 \wedge e_2 \wedge e_3; \tag{14}$$

this can be multiplied with the bivector

$$E := e_\infty \wedge e_0 = e_4 \wedge e_5 = e_4 e_5, \tag{15}$$

leading to the pseudo-scalar

$$I = I_3 E = e_1 \wedge e_2 \wedge e_3 \wedge e_\infty \wedge e_0 \tag{16}$$

in $G_{4,1}$, which is used for computing the dual of the multivectors in CGA.

For instance, the dual of a multivector $M \in G_{4,1}$ is given by

$$M^* := M I^{-1}; \tag{17}$$

from now on, we will use the symbol $^*$ to identify the dual of a multivector in this algebra.

Now, the principal geometric primitives used in this work are defined, and rigid transformations are introduced to give the background of the proposed algorithm of inverse kinematics.

### 2.3. Used Geometric Primitives

A *point* $x_c \in G_{4,1}$ is represented in the 5D conformal space by taking the linear combinations of some elements of its base:

$$\{e_1, e_2, e_3, e_\infty, e_0\}; \tag{18}$$

that is,

$$x_c = x_e + \frac{1}{2} x_e^2 e_\infty + e_0, \tag{19}$$

where $x_e = \alpha_1 e_1 + \alpha_2 e_2 + \alpha_3 e_3 \in \mathbb{R}^3$.

Note that $\mathbb{R}^3$ can be seen to be embedded in CGA.

We know that the equation of a *sphere* with its center at $p_e \in \mathbb{R}^3$ and radius $\rho \geq 0$ is given by

$$(x_e - p_e)^2 = \rho^2. \tag{20}$$

The sphere can be mapped to $G_{4,1}$ as

$$s = p_c - \frac{1}{2} \rho^2 e_\infty, \tag{21}$$

where $p_c \in G_{4,1}$ is the center of the sphere; (21) corresponds to the so-called inner product null space (IPNS) representation.

Note that when $\rho = 0$, we again obtain this point: that is, $s = p_c$. Taking into account the pseudo-scalar $I$, we can obtain the dual of the sphere $s^* = sI^{-1}$, which is represented as a 4-vector. This means that a sphere can be described by making the wedge product of four points that lie on it: namely,

$$s^* = x_{c_1} \wedge x_{c_2} \wedge x_{c_3} \wedge x_{c_4}. \tag{22}$$

If we make $x_{c_4} = e_\infty$, then we obtain a *plane*:

$$\pi^* = x_{c_1} \wedge x_{c_2} \wedge x_{c_3} \wedge e_\infty. \tag{23}$$

A plane is a sphere with an infinite radius.

The plane $\pi$ is also represented in IPNS form as follows:

$$\pi = n + d e_\infty, \tag{24}$$

where $n$ and $d$ are the normal vector and Hesse distance, respectively.

In $G_{4,1}$, we can also find a line that is normal to a plane $\pi$ as

$$L_{\pi}^* = (\pi^* \cdot E)/I, \tag{25}$$

where $E$ is the Minkowski plane and $I$ is the pseudo-scalar in CGA that we defined at the end of the above subsection.

Let $s_1$, $s_2 \in G_{4,1}$ be two spheres. We can obtain a *circle z* as the intersection of $s_1$ and $s_2$: that is,

$$z = s_1 \wedge s_2, \tag{26}$$

which is given in IPNS, and its dual form can be expressed by three points lying on the circle (similarly to the dual of a sphere) as

$$z^* = x_{c_1} \wedge x_{c_2} \wedge x_{c_3}. \tag{27}$$

Given the circle $z$ as in (26) and from (27), it follows that the plane $\pi_z^*$ in which the circle lies is given by

$$\pi_z^* = z^* \wedge e_{\infty}. \tag{28}$$

Now, we can have a line that passes through the center of the circle that is normal to the plane $\pi_z^*$ as

$$L_z^* = z \wedge e_{\infty}. \tag{29}$$

These properties of the primitives will be used as a part of the proposed algorithm of inverse kinematics.

Similar to the case of planes, *lines* can be defined by changing $x_{c3} = e_{\infty}$ in (27), which gives

$$L^* = x_{c_1} \wedge x_{c_2} \wedge e_{\infty}. \tag{30}$$

A line can also be expressed as

$$L = \mathbf{n}I_3 + e_{\infty}\mathbf{m}I_3 \tag{31}$$

in the standard IPNS form, where $\mathbf{n}$ and $\mathbf{m}$ are the line orientation and moment, respectively.

Finally, the *point pair* is used as two possible solutions of a joint that is given by

$$P_p^* = p_1 \wedge p_2 \tag{32}$$

by using the wedge product of the two points $p_1$ and $p_2$. Alternatively, we can also get the point pair as the intersection of a line and a sphere, a line and a circle, or a circle and a sphere.

Given $P_p^*$, we can compute $p_1$, $p_2$ as

$$p_{1,2} = \frac{P_p^* \pm \left(P_p^* \cdot P_p^*\right)^{1/2}}{P_p^* \cdot e_{\infty}}. \tag{33}$$

*2.4. Translation, Rotation, and Motor*

Rigid transformations can be expressed in CGA by composing plane reflections. For any geometric entity, say $Q$, the reflection with respect to a plane $\pi$ is given by

$$Q' = \pi Q \pi^{-1}. \tag{34}$$

The translation can be seen as the reflection with respect to the planes $\pi_1 = n + 0e_{\infty}$, $\pi_2 = n + de_{\infty}$. It follows that the translation is given by

$$Q' = (\pi_2 \pi_1)Q(\pi_1^{-1}\pi_2^{-1}) = T_a Q \widetilde{T}_a, \tag{35}$$

where

$$T_a = (n + de_\infty)n = e^{-\frac{a}{2}e_\infty},\tag{36}$$

$a = 2dn$, $\|n\| = 1$, and $\widetilde{T}_a$ is called the reversion.

Similarly, the rotation can be described as the composition of two reflections with respect to $\pi_1 = n_1$, $\pi_2 = n_2$, which is

$$Q' = (\pi_2\pi_1)Q(\pi_1^{-1}\pi_2^{-1}) = R_\theta Q\widetilde{R}_\theta,\tag{37}$$

where

$$R_\theta = \pi_2\pi_1 = n_2 \cdot n_1 + n_2 \wedge n_1 = cos(\frac{\theta}{2}) - sin(\frac{\theta}{2})l = e^{-\frac{\theta}{2}l},\tag{38}$$

$l = n_1 \wedge n_2$, and $\theta$ is double the angle between $\pi_2$ and $\pi_1$.

A rigid-body motion that includes translation and rotation is described by a motor (also called a displacement versor) $Mot = TR$. Thus, the rigid body motion of $Q$ is described by

$$Q' = MotQ\widetilde{Mot}.\tag{39}$$

## 3. Inverse Kinematics Methodology

The human upper limb is described as a system of three segments: the upper arm, the forearm, and the hand; it includes seven degrees of freedom. This configuration is basic to emulate human movements and is used in this paper. We present a summary of the algorithm proposed in Algorithm 1. To explain the IK of a humanoid robot arm, an anthropomorphic 7-DoF arm is used; the configuration of the real robot to implement the IK is represented in Figure 1.

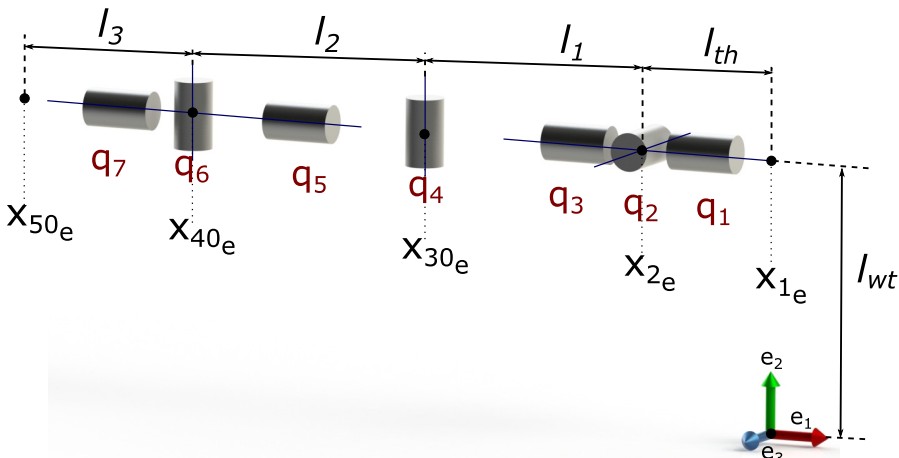

**Figure 1.** Initial configuration of the 7-DoF arm.

Usually, the desired trajectory of the hand of a humanoid robot is given to satisfy complex tasks, i.e., around an object, to achieve the desired position, etc. Then, the amount of joint rotation of the wrist, elbow, and shoulder must be found with this information.

The desired position and attitude of the gripper at the hand are given in the form:

$$x_g, y_g, z_g\tag{40}$$

for the $e_1$, $e_2$, and $e_3$ axes, and the pitch, yaw, and roll are

$$\alpha_g, \beta_g, \gamma_g\tag{41}$$

with respect to the frame coordinates fixed to the center of the torso. We define $l_1$ as the distance between the shoulder and the elbow joint, $l_2$ as the distance between the elbow and the wrist, and finally, $l_3$ as the distance from the wrist to the gripper, as shown in

Figure 1, where $l_{wt}$ and $l_{th}$ are the distances between the shoulder frame with respect to the fixed torso frame in the $e_2$ and $e_1$ directions, respectively.

---

**Algorithm 1** Inverse Kinematics for 7-DoF Manipulator.

---

**Input:** $x_g, y_g, z_g, \alpha_g, \beta_g, \gamma_g, l_1, l_2, l_3, l_{wt}, l_{th}$
**Output:** $q_1, q_2, q_3, q_4, q_5, a_6, q_7$

$\quad x_1 \leftarrow 0e_1 + l_{wt}e_2 + 0e_3 + \frac{1}{2}(0e_1 + l_{wt}e_2 + 0e_3)^2 e_\infty + e_0$

$\quad x_2 \leftarrow -l_{th}e_1 + l_{wt}e_2 + 0e_3 + \frac{1}{2}(-l_{th}e_1 + l_{wt}e_2 + 0e_3)^2 e_\infty + e_0$

$\quad x_3 \leftarrow \frac{P_{pe}^* + \sqrt{P_{pe}^* \cdot P_{pe}^*}}{P_{pe}^* \cdot e_\infty}$

$\quad x_4 \leftarrow \frac{P_{pw}^* + \sqrt{P_{pw}^* \cdot P_{pw}^*}}{P_{pw}^* \cdot e_\infty}$

$\quad x_p \leftarrow M_g w \tilde{M}_g$

$\qquad$ Links:

$Link_0 \leftarrow e_\infty \wedge x_1 \wedge x_2$

$Link_1 \leftarrow e_\infty \wedge x_2 \wedge x_3$

$Link_2 \leftarrow e_\infty \wedge x_3 \wedge x_4$

$Link_3 \leftarrow e_\infty \wedge x_4 \wedge x_p$

$\qquad$ Joints:

$q_1 \leftarrow -\arccos\left(\frac{\sqrt{P_a^* \cdot P_{31}^*}}{|P_a^*||P_{31}^*|}\right)$

$q_2 \leftarrow -\arccos\left(\frac{\sqrt{Link_1 \cdot Link_0}}{|Link_1||Link_0|}\right)$

$q_3 \leftarrow \arccos\left(\frac{\sqrt{P_{q3r}^* \cdot P_{q3}^*}}{|P_{q3r}^*||P_{q3}^*|}\right)$

$q_4 \leftarrow \arccos\left(\frac{\sqrt{Link_2 \cdot Link_1}}{|Link_2||Link_1|}\right)$

$q_5 \leftarrow \arccos\left(\frac{\sqrt{P_{q5r}^* \cdot P_{q5}^*}}{|P_{q5r}^*||P_{q5}^*|}\right)$

$q_6 \leftarrow \arccos\left(\frac{\sqrt{Link_3 \cdot Link_2}}{|Link_3||Link_2|}\right)$

$q_7 \leftarrow \arccos\left(\frac{\sqrt{P_{q7}^* \cdot P_{xp}^*}}{|P_{q7}^*||P_{xp}^*|}\right)$

---

Development of the algorithm is separated into the following two sections: In the first section, we define the geometric entities that will be used in the algorithm: in particular, those used to compute the joint positions in CGA. In the second section, we use the definition of the inner product to find the angles of each joint. The proposed method is based on our previous works [1,12] given that the first four joint angles are calculated in the same way for a 5-DoF robot arm or 6-DoF leg.

### 3.1. Defining the Geometric Entity References

First, we define the Euclidean positions representing the initial joint positions that are shown in Table 1.

**Table 1.** Joint positions according to Figure 1.

| Reference Point | $x$, $y$, and $z$ Position | Description |
| :---: | :---: | :---: |
| $x_{1_e}$ | $(0, l_{wt}, 0)$ | Center of the torso |
| $x_{2_e}$ | $(-l_{th}, l_{wt}, 0)$ | Shoulder initial position |
| $x_{30_e}$ | $(-(l_{th} + l_1), l_{wt}, 0)$ | Elbow initial position |
| $x_{40_e}$ | $(-(l_{th} + l_1 + l_2), l_{wt}, 0)$ | Wrist initial position |
| $x_{50_e}$ | $(-(l_{th} + l_1 + l_2 + l_3), l_{wt}, 0)$ | Gripper initial position |

Since $x_{1_e}$ and $x_{2_e}$ are fixed with respect to the world coordinate system, their conformal representations are given by

$$x_1 = 0e_1 + l_{wt}e_2 + 0e_3 + \frac{1}{2}(0e_1 + l_{wt}e_2 + 0e_3)^2 e_\infty + e_0, \tag{42}$$

and similarly for

$$x_2 = -l_{th}e_1 + l_{wt}e_2 + 0e_3 + \frac{1}{2}(-l_{th}e_1 + l_{wt}e_2 + 0e_3)^2 e_\infty + e_0. \tag{43}$$

Three rotors describing the desired pitch, yaw, and roll of the gripper are created in the form:

$$\begin{aligned} R_{x_g} &= e^{-\frac{1}{2}\alpha_g e_{23}}, \\ R_{y_g} &= e^{-\frac{1}{2}\beta_g e_{31}}, \\ R_{z_g} &= e^{-\frac{1}{2}\gamma_g e_{12}}, \end{aligned} \tag{44}$$

where $e_{23}$, $e_{31}$ and $e_{12}$ are bivectors.

Thus, the general rotor describing the entire desired attitude is given in the form:

$$R_g = R_{x_g} R_{y_g} R_{z_g}. \tag{45}$$

Then, with the desired Euclidean coordinates, a translator is formed:

$$T_g = e^{-\frac{1}{2}(x_g e_1 + y_g e_2 + z_g e_3)e_\infty}, \tag{46}$$

where $x_g e_1 + y_g e_2 + z_g e_3 \in \mathbb{R}^3$.

Using the rotor (45) and translator (46), a motor describing the desired pose of the gripper in terms of the fixed torso frame is given in the form:

$$M_g = T_g R_g. \tag{47}$$

It follows that the pose of the gripper is given by

$$x_p = M_g w \tilde{M}_g, \tag{48}$$

where $w$ is a conformal point of (42) representing the origin of the world coordinate system.

Now, it remains to calculate the joint positions $x_3$ and $x_4$; to this end, several geometric entities are defined.

First, a sphere and a line are created about the zero conformal point $e_0$ using the distance $l_3$ between the wrist and the gripper as follows:

$$\begin{aligned} S_0 &= e_0 - \frac{1}{2}l_3^2 e_\infty, \\ L_0 &= e_2 e_3. \end{aligned} \tag{49}$$

Then, the sphere and the line given in (49) are translated and rotated by the motor $M_g$ to the desired pose of the end-effector:

$$\begin{aligned} S_{ref} &= M_g S_0 \tilde{M}_g, \\ L_{ref} &= M_g L_0 \tilde{M}_g \end{aligned} \tag{50}$$

such that the center of this translated sphere $S_0$ is the gripper position of $x_p$, and the line $L_0$ meets this point.

The sphere and the line defined in (50) are intersected to obtain a point pair that represents the two possible solutions for the conformal positions of the wrist and are given by

$$P_{pw} = S_{ref} \wedge L_{ref}. \tag{51}$$

The point pair can be separated using the equation

$$x_4 = \frac{P_{pw}^* + \sqrt{P_{pw}^* \cdot P_{pw}^*}}{P_{pw}^* \cdot e_\infty}, \tag{52}$$

which gives the wrist a conformal joint position.

An illustration of how to get $x_4$ when implemented is given in Figure 2.

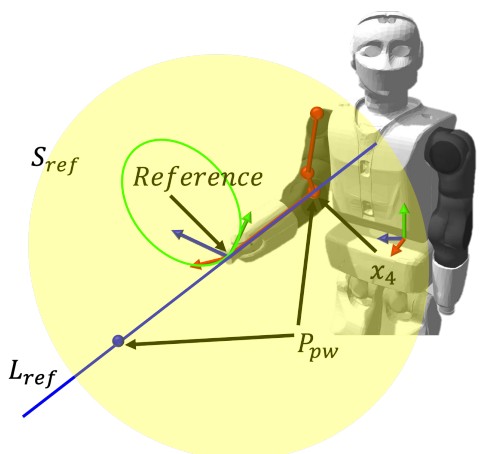

**Figure 2.** Geometric entities used to find $x_4$.

Now, two spheres have been defined: one with its origin at the point $x_2$ after mapping $x_{2_e}$ (Table 1) in CGA and the second one with its center at the point $x_4$ obtained from (52); the radii of the spheres are the distances $l_1$ and $l_2$, respectively, as shown in Figure 1, and are defined as

$$S_1 = x_2 - \frac{1}{2}l_1^2 e_\infty,$$
$$S_2 = x_4 - \frac{1}{2}l_3^2 e_\infty. \tag{53}$$

The intersection of the two spheres from (53) gives a circle that contains all the possible positions of the elbow and is given by

$$Z = S_1 \wedge S_2; \tag{54}$$

this circle will be intersected with a plane to find the elbow position. This circle contains an infinite number of points to place the robot's elbow; when intersecting it with the plane, we find a pair of points, and it is necessary to select one of them to define the configuration of the arm, If we want to find another configuration, it is possible to take the initial point and rotate this point with respect to the center of the circle and find a new one. With this procedure, it is possible to take advantage of the redundancy of the seven degrees of freedom of the arm: for example, in obstacle avoidance tasks. The development of these operations can be found in our previous work [12].

The plane is constructed as follows:

As shown in Figure 3, a line normal to the circle of (54) is calculated in the form

$$L_z^* = Z \wedge e_\infty; \tag{55}$$

this line can be used to find a plane in which the arm lies and, as future work, could help to design systems for obstacle and self-collision avoidance.

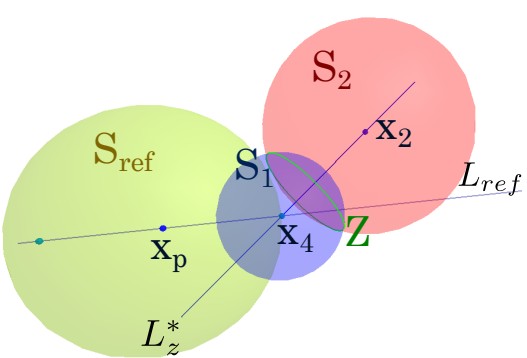

**Figure 3.** Intersections of geometric entities to calculate $x_4$ and all the possible positions for $x_3$ (which lies in a circle).

Next, a reference line parallel to the $e_1$ axis is translated to the point $x_2$; this is given using a translator defined as follows:

$$T_{x_2} = e^{-\frac{1}{2}(-l_{th}e_1 + l_{wt}e_2)e_\infty},$$ (56)

and the reference line is given by

$$L_{e_1}^* = T_{x_2} E e_1 \tilde{T}_{x_2},$$ (57)

where $-l_{th}e_1 + l_{wt}e_2 \in \mathbb{R}^3$ and $E = e_\infty \wedge e_0$.

Now, using the lines $L_z^*$ and $L_{e_1}^*$, the plane is defined as

$$P_a^* = \left(L_z^* \cdot (e_\infty \wedge e_0)\right) \wedge L_{e_1}^*.$$ (58)

Then, a point pair representing the two possible positions of the elbow can be found: to do this, the circle in (54) is intersected with the plane of (58) using the equation

$$P_{pe} = Z \wedge P_a^*.$$ (59)

As before, the point pair is separated, and the elbow point is given by

$$x_3 = \frac{P_{pe}^* + \sqrt{P_{pe}^* \cdot P_{pe}^*}}{P_{pe}^* \cdot e_\infty}.$$ (60)

This procedure is shown in Figure 4.

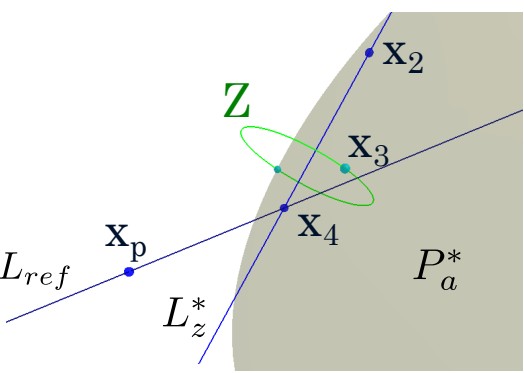

**Figure 4.** Intersection of the plane $P_a^*$ with $Z$ to get the two possible positions of $x_3$.

Finally, using the previously found points, four lines representing the links of the arm are calculated; these lines are shown in Figure 5 and are given by

$$
\begin{aligned}
Link_0 &= e_\infty \wedge x_1 \wedge x_2, \\
Link_1 &= e_\infty \wedge x_2 \wedge x_3, \\
Link_2 &= e_\infty \wedge x_3 \wedge x_4, \\
Link_3 &= e_\infty \wedge x_4 \wedge x_p.
\end{aligned}
\tag{61}
$$

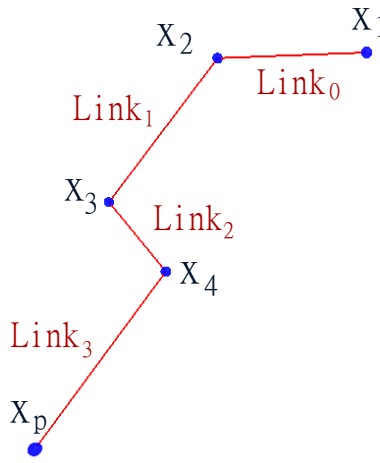

**Figure 5.** The lines and points defining the configuration that are needed in order to reach the desired pose.

These lines will be used in the next subsection to find some of the joint angles by taking into account their inner product.

### 3.2. Computing the Joint Values of the Arm

With the points and geometric entities defined, the angles of each joint are calculated: the first four are calculated as in [1,12].

First, to find the rotation of $q_1$, a plane parallel to the axes $e_3$ and $e_1$ is defined; then, using a translator, this plane is translated to the point $x_2$ as follows:

$$
P_{31}^* = T_{x_2}\left( (e_\infty \wedge e_0)e_3 e_1 \right) \tilde{T}_{x_2}.
\tag{62}
$$

The amount of rotation is given by the angle between the planes of (58) and (62) and is calculated by

$$
q_1 = -\arccos\left( \frac{\sqrt{P_a^* \cdot P_{31}^*}}{|P_a^*||P_{31}^*|} \right).
\tag{63}
$$

The joint value $q_2$ is defined as the angle between the lines $Link_0$ and $Link_1$ of (61) and is obtained as

$$
q_2 = -\arccos\left( \frac{\sqrt{Link_1 \cdot Link_0}}{|Link_1||Link_0|} \right).
\tag{64}
$$

In order to find $q_3$, two planes are defined as

$$
\begin{aligned}
P_{q3r}^* &= e_\infty \wedge x_1 \wedge x_2 \wedge x_3, \\
P_{q3}^* &= e_\infty \wedge x_3 \wedge x_2 \wedge x_4,
\end{aligned}
\tag{65}
$$

and $q_3$ is defined as the angle between the planes defined in (65) and is calculated as

$$q_3 = \arccos\left(\frac{\sqrt{P^*_{q3r} \cdot P^*_{q3}}}{|P^*_{q3r}||P^*_{q3}|}\right). \tag{66}$$

The angle $q_4$ is defined as the angle between the lines $Link_1$ and $Link_2$ defined in (61); this angle is obtained as

$$q_4 = \arccos\left(\frac{\sqrt{Link_2 \cdot Link_1}}{|Link_2||Link_1|}\right). \tag{67}$$

In the same way, the joint value $q_6$ is the angle between the lines $Link_2$ and $Link_3$ in (61) and is given by

$$q_6 = \arccos\left(\frac{\sqrt{Link_3 \cdot Link_2}}{|Link_3||Link_2|}\right). \tag{68}$$

The angles $q_5$ and $q_7$ are difficult to compute because of the redundancy of the 7-DoF robotic arm; for that reason, in conventional approaches such as matrix representation, direct kinematics are used to know where the arm is based on previously computed angles, and this leads to many computations, which costs time to compute the inverse kinematics in real-time. To avoid the need to use direct kinematics in this approach, four planes are defined as geometric restrictions to compute the two angles: these planes are shown in Figures 6 and 7.

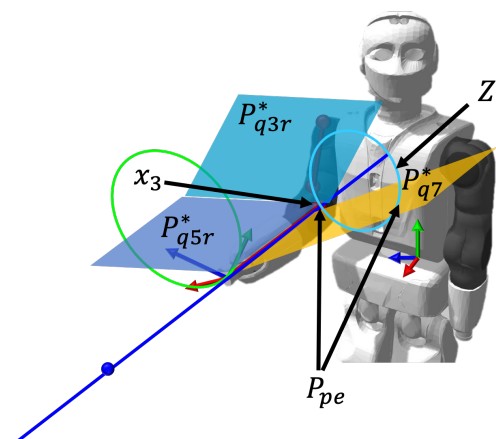

**Figure 6.** Geometric entities used in order to compute $q_5$.

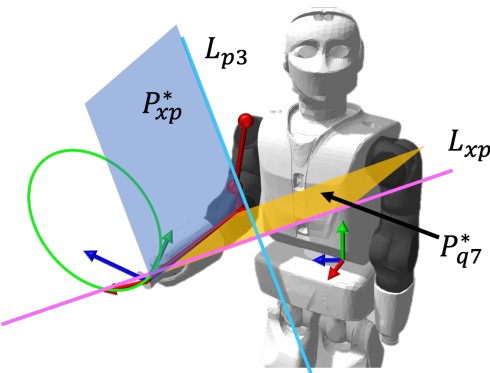

**Figure 7.** Geometric entities used in order to compute $q_7$.

First, using the plane $P^*_{q3}$ defined in (65), a line normal to this plane is calculated; then, this line is translated to the calculated point $x_4$ as follows:

$$L^*_{P3} = T_{x_4}((P^*_{q3} \cdot E)/I)\tilde{T}_{x_4}. \tag{69}$$

where $T_{x_4}$ is given similar to (56).

Using the computed point $x_3$ and the goal position $x_p$, two planes are defined as follows:

$$\begin{aligned} P_{q5}^* &= L_{P3}^* \wedge x_3, \\ P_{x_p}^* &= L_{P3}^* \wedge x_p. \end{aligned} \tag{70}$$

After that, a plane is defined using the calculated points and the goal position as

$$P_{q5r}^* = e_\infty \wedge x_3 \wedge x_4 \wedge x_p. \tag{71}$$

The angle $q_5$ is defined as the angle between the planes $P_{q5}^*$ and $P_{q5r}^*$ as

$$q_5 = \arccos\left(\frac{\sqrt{P_{q5r}^* \cdot P_{q5}^*}}{|P_{q5r}^*||P_{q5}^*|}\right). \tag{72}$$

Finally, to compute the joint angle $q_7$, the next steps are followed. First, we obtain the normal line to the plane $P_{q5r}^*$, and as in (69), this line is translated to the goal position $x_p$ as follows:

$$L_{xp}^* = T_g\left((P_{q5r}^* \cdot E)/I\right)\tilde{T}_g, \tag{73}$$

where $T_g$ is the translator given in (46).

Now, using this line $L_{xp}^*$ and the point $x_4$, a plane is defined as

$$P_{q7}^* = L_{xp}^* \wedge x_4. \tag{74}$$

The joint value $q_7$ is defined as the angle between the planes $P_{q7}^*$ and $P_{xp}^*$ and is given by

$$q_7 = \arccos\left(\frac{\sqrt{P_{q7}^* \cdot P_{xp}^*}}{|P_{q7}^*||P_{xp}^*|}\right). \tag{75}$$

Therefore, we have solved the inverse kinematics problem, where given a desired pose

$$x_g, y_g, z_g, \alpha_g, \beta_g, \gamma_g, \tag{76}$$

we find the appropriate joint angles in order for the end-effector of the 7-DoF robotic arm to move to that desired pose.

## 4. Results

*Simulations*

To demonstrate the effectiveness of the proposed method, several experiments were performed. One of them uses the time-dependent reference given by the desired position and attitude of the gripper at the hand:

$$\begin{aligned} x_g &= -0.1 - 0.08\cos(0.2t), \\ y_g &= 0.1 + 0.8\sin(0.2t), \\ z_g &= 0.35, \end{aligned} \tag{77}$$

and the pitch, yaw, and roll are

$$\alpha_g = 0, \ \beta_g = -\frac{\pi}{2}, \ \gamma_g = 0, \tag{78}$$

where the dimensions of the position are given in meters, and for the attitude, they are given in radians. The simulation was done using Clucalc [13,14], and the results of the joint angles are shown in Figure 8.

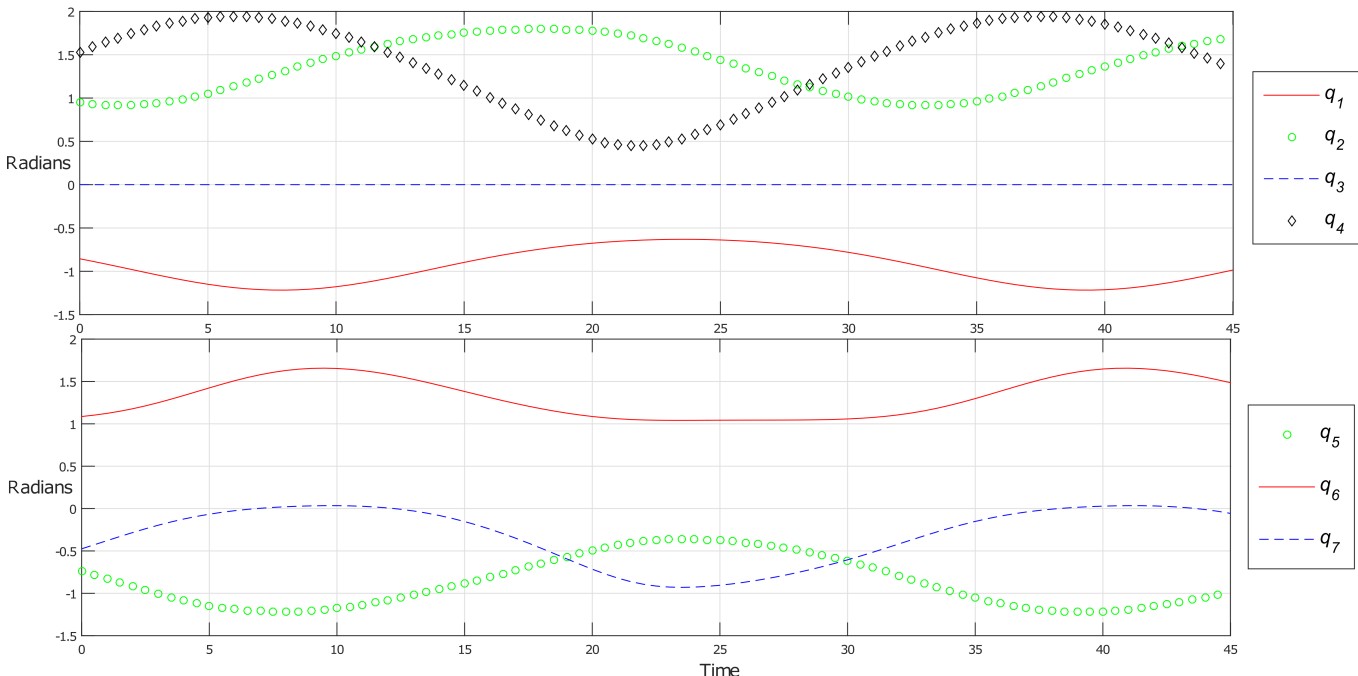

**Figure 8.** Inverse kinematics solution of a 7-DoF robotic arm.

Finally, for the sake of comparison, we implemented our algorithm in C++ language and compared it with the state-of-the-art algorithms [15,16]. The benchmark was performed on a PC running Ubuntu 14.04-LTS with an Intel Core i5-3210M @ 2.50 GHz processor and 4 GB of RAM.

The test setup consisted of generating 1000 random poses composed of the position and orientation of the end-effector. The different poses served as input for the algorithm proposed in [15,16]. The arms from the Mex-One robot, REEM-C, and the PR2 were selected to test the different algorithms. The results are shown in Table 2.

The solve rate of our algorithm shows improvement compared to [15] and has similar performance as [16]. Nevertheless, our algorithm outperforms in the time for computing a solution compared to using inverse kinematics.

**Table 2.** A comparison with the state-of-the-art.

| Robot | Position/Rotation Error | KDLSolve Rate (%) | Avg Time (ms) | TRAC-IKSolve Rate (%) | Avg Time (ms) | FIKASolve Rate (%) | Avg Time (ms) |
|---|---|---|---|---|---|---|---|
| Mex-One | $1 \times 10^{-6}/1 \times 10^{-6}$ | 79.8 | 1.57704 | **98.9** | 0.88694 | 94.9 | **0.14862** |
| REEM-C | $1 \times 10^{-6}/1 \times 10^{-6}$ | 81.3 | 1.47647 | **99.8** | 0.66928 | 95.6 | **0.15064** |
| PR2 | $1 \times 10^{-6}/1 \times 10^{-6}$ | 81.7 | 1.46981 | **99.7** | 0.68159 | 92.9 | **0.16407** |

## 5. Real-Time Results

Several experiments were performed to demonstrate the effectiveness of the proposed method. As explained before, a simulation was done using the time-dependent reference given by (77). Note that all the position dimensions are in meters, and the orientation goals are in radians.

Using the pose reference, a simulation was done using Clucalc [13,14]; the joint values obtained from the simulation are shown in the Figure 8.

Finally, online implementation was done in a full-sized humanoid platform. Seven DoFs configure the humanoid arm, each with a Dynamixel series servomotor. Three printed fingers conform to the gripper, each wired to one DC motor. Composite images of the experiment are shown in the Figure 9. The complete video of an example of the

experiments can be found in the Supplementary Materials. With this real-time experiment, we demonstrate that our algorithm can be implemented in robot platforms. First, we designed a desired trajectory for the end effector, a circle inside the plane formed by $e_1$ and $e_2$, with different radius and a distance in $e_3$ direction such that it lies in the workspace of the robot arm. The home position of our arm is the same as presented in Figure 1, meaning the arm is fully stretched. Thus, we described the circle trajectory by restricting the orientation of the robot hand and moving the $e_1$ and $e_2$ coordinates according to the circle equation discretized up to a delta time of 0.1s. The inverse kinematics is computed at each time step, and the joint angles calculated are commanded to the Dynamixcel motors.

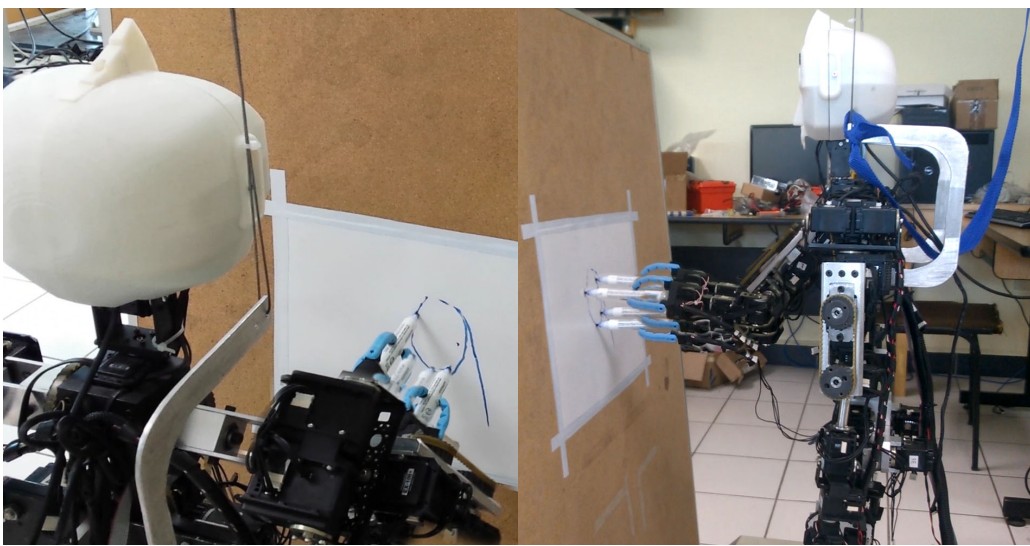

**Figure 9.** Experimental results of the inverse kinematic algorithm implemented in the Mex-One humanoid robot prototype.

## 6. Conclusions

In this paper, a novel method to solve the inverse kinematics for seven degrees of freedom for a redundant robotic arm using the conformal geometric algebra framework is proposed. Taking advantage of geometric entities defined in this algebra, it was possible to find a solution to the problem without using forward kinematics, numerical solutions, or many matrix multiplications.

This property allowed us to decrease the amount of calculations needed to reach the desired goal. A comparison with the state-of-the-art algorithms was presented and demonstrated the superior performance of the algorithm discussed here.

As future work, the addition of geometric restrictions to avoid joint limits will be considered: including algorithms to avoid self-collision and for obstacle avoidance.

Moreover, an online implementation in a real robot and a complexity analysis of the proposed algorithm must be done.

**Supplementary Materials:** The following supporting information can be downloaded at: https://www.mdpi.com/article/10.3390/machines12010078/s1.

**Author Contributions:** Conceptualization, O.C.-E. and L.C.-M.; methodology, O.C.-E.; software, L.C.-M.; validation, O.C.-E., L.C.-M. and M.D.-R.; formal analysis, O.C.-E.; investigation, O.C.-E. and L.C.-M.; resources, M.D.-R.; data curation, L.C.-M.; writing—original draft preparation, O.C.-E., L.C.-M. and M.D.-R.; writing—review and editing, O.C.-E., L.C.-M. and M.D.-R.; visualization, O.C.-E., L.C.-M. and M.D.-R.; supervision, M.D.-R.; project administration, O.C.-E. All authors have read and agreed to the published version of the manuscript.

**Funding:** This research received no external funding.

**Data Availability Statement:** The data presented in this study are available in the article.

**Conflicts of Interest:** Author Leobardo Campos-Macías was employed by the company Intel Corporation. The remaining authors declare that the research was conducted in the absence of any commercial or financial relationships that could be construed as a potential conflict of interest.

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
