# Peer review of "FIKA: A Conformal Geometric Algebra Approach to a Fast Inverse Kinematics Algorithm for an Anthropomorphic Robotic Arm"

_machines, doi:10.3390/machines12010078_

Round 1

Reviewer 1 Report

Comments and Suggestions for Authors

This paper proposes a geometric approach to solve the inverse kinematics for an anthropomorphic robotic arm with seven DOFs, along with demonstrations by simulations and experiments. The analytical derivations are well described and demonstrated. However, the following are some minor points that should be further addressed.

Major concerns

-The comparative study is merely conducted through simulations, and it is unclear how the proposed controller works in the experimental rig. Comparative studies against the existing approaches should also be performed experimentally.

Minor comments

-Video demonstrations would be appreciated to facilitate readers.

-“Hybrid adaptive disturbance rejection control for inflatable robotic arms” may be useful for reference.

Comments on the Quality of English Language

Good language.

Author Response

-The comparative study is merely conducted through simulations, and it is unclear how the proposed controller works in the experimental rig. Comparative studies against the existing approaches should also be performed experimentally.

Thanks for the comments. We demonstrate our algorithm performance in the MexOne prototype platform. The paper aims to generate the joint configuration of a redundant robot arm using analytical methodologies provided by the conformal geometric algebra framework. Thus, we are not evaluating the low-level joint controller. Nonetheless, we include state-of-the-art comparisons in Section 4.

-Video demonstrations would be appreciated to facilitate readers.

Thanks, we will include the video in a youtube link once the paper is accepted.

-“Hybrid adaptive disturbance rejection control for inflatable robotic arms” may be useful for reference.

This reference is not relevant since the paper's scope is not the controller's design.

Reviewer 2 Report

Comments and Suggestions for Authors

Dear authors,

The work presents an approach to solving the inverse kinematics problem for a specified anthropomorphic robotic arm with seven degrees of freedom, using a modern mathematical apparatus: conformal geometric algebra. The article builds upon the authors' previous work in the field, which I consider to be a good practice. Although section 2 - Mathematical Background, introduces a well-known mathematical apparatus, I think the section helpfully presents its application in robotics.

My main recommendations are related to improving: the description of the algorithm; the presentation of the experiment; the analysis of the algorithm, the experiment and the results. The aim is to describe the concepts, objectives and results of the research as clearly as possible, which will improve the impact of the article.

Please find my comments below:

1.      The literature references can be expanded by adding articles on inverse kinematics in refereed journals from the past 10 years. It would be good to include a DOI for every source where available. Information on literary sources [12] and [14] is incomplete.

2.      I recommend specifying the aim of the research at the end of the Introduction.

3.      Only part of the formulas are numbered. It is better to number all the formulas or at least the ones from the experiment so they can be commented on.

4.      Check that all notations in the formulas are clearly defined. For example, what are M*, L*… ?

5.      “Algorithm” is in the title, so the algorithm must be clearly defined. I suggest including either a step-by-step description, a pseudocode or a flowchart. Perhaps: 1. Geometric primitives and operators for working with them are created; 2. The parameters of the robot are entered; 5. Multiple solutions of the inverse problem are found using a section of two spheres; etc.

6.      Since the robot is redundant, in the general case many solutions of the inverse kinematics are obtained. Explain how a single solution is chosen. Does choosing a single solution with the proposed 4 planes optimize anything?

7.      Give information about the practical experiment. How did you determine the position and orientation of the robot relative to the sheet of paper, so as to fulfill the trajectory (circle) from the formula in 4.1 - Simulations? What are the hand’s joint limitations? What is the Home position of the hand? It is not clear if the experiment starts from the Home position and goes to draw a circle or from an arbitrary point of the circle? If possible, measure the parameters of the resulting circle.

8.      In practice, there is no Discussion. I suggest that the text written in the current Discussion section goes to the Conclusion. In the Conclusion, it is good to indicate how your results can be used by other researchers. I understand that in the template of this journal there is no mandatory conclusion, but still, every article would benefit from a Conclusion.

9.      In the Discussion section, assess the advantages and disadvantages of your algorithm. In which similar cases would it be applicable? What are the limitations of the algorithm? Discuss the experiment.

10.  Comment on what you would do in the following situations: A point may have an IK solution but be outside the joint constraints. In some cases, a point may be reachable by only one singular configuration (with the spheres touching). As far as I understand, joint constraints are not taken into account when solving the IK. It is well-known that, depending on the joint constraints, such a robot can have two types of solutions: left-handed and right-handed. No check is made whether the target point is within the workspace.

Best regards,

Reviewer

Author Response

  1. The literature references can be expanded by adding articles on inverse kinematics in refereed journals from the past 10 years. It would be good to include a DOI for every source where available. Information on literary sources [12] and [14] is incomplete.

Thank you for your recommendations, we have corrected 12 and 14. We also have included a survey paper for general inverse kinematics algorithms.

  1. I recommend specifying the aim of the research at the end of the Introduction.

We included in the paper: “The principal aim of this research work is to develop an inverse kinematics algorithm using Conformal Geometric Algebra that allows us to extend the previous work we have been doing to redundant robotic arms with 7 degrees of freedom. With our proposal, the calculation time is reduced by exploiting the properties of the mathematical framework with which we are working. By using this algorithm, we hope to improve the performance of this type of manipulator arms in real time”.

  1. Only part of the formulas are numbered. It is better to number all the formulas or at least the ones from the experiment so they can be commented on.

Thanks, we numbered all formulas.

  1. Check that all notations in the formulas are clearly defined. For example, what are M*, L*… ?

 It was specified that the symbol “*” will be used to represent the dual of a multivector in equation 17. We included in the paper.

  1. “Algorithm” is in the title, so the algorithm must be clearly defined. I suggest including either a step-by-step description, a pseudocode or a flowchart. Perhaps: 1. Geometric primitives and operators for working with them are created; 2. The parameters of the robot are entered; … 5. Multiple solutions of the inverse problem are found using a section of two spheres; etc.

We included a pseudocode of the algorithm in Algorith 1. With inputs and outputs, and a draft for points, lines and angles calculations.

  1. Since the robot is redundant, in the general case many solutions of the inverse kinematics are obtained. Explain how a single solution is chosen. Does choosing a single solution with the proposed 4 planes optimize anything?

We included in the paper:

“The intersection of the two spheres of~\eqref{eq:9} gives a circle which

contains all the possible positions of the elbow, and is given by

\begin{equation}\label{eq:10}

Z=S_1\wedge S_2,

\end{equation}

this circle will be intersected with a plane to find the elbow position. This circle  contains an infinite number of points to place the robot's elbow, when intersecting it with the plane we find a pair of points and it is necessary to select one of them to define the configuration of the arm, If we want to find another configuration, it is possible to take the initial point and rotate this point with respect to the center of the circle and find a new one. With this procedure it is possible to take advantage of the redundancy of the 7 degrees of freedom arm, for example in obstacle avoidance tasks. The development of these operations can be found in our previous work \cite{oscaravoid}.”

  1. Give information about the practical experiment. How did you determine the position and orientation of the robot relative to the sheet of paper, so as to fulfill the trajectory (circle) from the formula in 4.1 - Simulations? What are the hand’s joint limitations? What is the Home position of the hand? It is not clear if the experiment starts from the Home position and goes to draw a circle or from an arbitrary point of the circle? If possible, measure the parameters of the resulting circle.

We included in the paper:

“Finally, online implementation was done in a full-sized humanoid platform. Seven DoFs configure the humanoid arm, each with a Dynamixel series servomotor. Three printed fingers conform to the gripper, each wired to one DC motor. Composite images of the experiment are shown in the figure~\ref{fig:CircleExper}. With this real-time experiment, we demonstrate that our algorithm can be implemented in robot platforms. First, we designed a desired trajectory for the end effector, a circle inside the plane formed by $e_1$ and $e_2$, with different radius and a distance in $e_3$ direction such that it lies in the workspace of the robot arm. The home position of our arm is the same as presented in figure~\ref{fig:1}, meaning the arm is fully stretched. Thus, we described the circle trajectory by restricting the orientation of the robot hand and moving the $e_1$ and $e_2$ coordinates according to the circle equation discretized up to a delta time of 0.1s. The inverse kinematics is computed at each time step, and the joint angles calculated are commanded to the Dynamixcel motors.”

  1. In practice, there is no Discussion. I suggest that the text written in the current Discussion section goes to the Conclusion. In the Conclusion, it is good to indicate how your results can be used by other researchers. I understand that in the template of this journal there is no mandatory conclusion, but still, every article would benefit from a Conclusion.

 The section name was changed for Conclusion

  1. In the Discussion section, assess the advantages and disadvantages of your algorithm. In which similar cases would it be applicable? What are the limitations of the algorithm? Discuss the experiment.

Thanks, discussion of the experiment has been added.

  1. Comment on what you would do in the following situations: A point may have an IK solution but be outside the joint constraints. In some cases, a point may be reachable by only one singular configuration (with the spheres touching). As far as I understand, joint constraints are not taken into account when solving the IK. It is well-known that, depending on the joint constraints, such a robot can have two types of solutions: left-handed and right-handed. No check is made whether the target point is within the workspace.

We are not considering these situations, but those are improvements opportunities that are being actively searching.